# Lip Lifting Efficacy of Hyaluronic Acid Filler Injections: A Quantitative Assessment Using 3-Dimensional Photography

**DOI:** 10.3390/jcm11154554

**Published:** 2022-08-04

**Authors:** Nark-Kyoung Rho, Boncheol Leo Goo, Seong-Jae Youn, Chong-Hyun Won, Kwang-Ho Han

**Affiliations:** 1Leaders Aesthetic Laser & Cosmetic Surgery Center, Seoul 06014, Korea; 2Department of Dermatology, Sungkyunkwan University School of Medicine, Seoul 06351, Korea; 3Naeum Dermatology and Aesthetic Clinic, Seoul 04512, Korea; 4Department of Dermatology, University of Ulsan College of Medicine, Seoul 05505, Korea; 5Nature Dermatology Clinic, Seoul 06055, Korea

**Keywords:** lip augmentation, lip projection, lip lift, columella-labial angle, nasolabial angle, 3-D surface imaging, hyaluronic acid, injectable fillers, cross-linking, lifting capacity

## Abstract

The study aimed to compare the volume enhancement and the lifting capacity of two different hyaluronic acid gels for lip injection. Thirty-six Korean female patients were randomized into two groups according to the cross-linking degree of the hyaluronic acid filler injected. Using a fixed injection protocol, patients were injected with 1 mL of hyaluronic acid filler in the lips and followed up at four and 12 weeks after injection. Lip volume, lip projection, and columella–labial angle were measured using a 3-dimensional imaging system at each time point. Follow-up values were compared with baseline. Compared with pre-treatment values, there was a statistically significant increase in mean lip volume and lip projection at four and 12 weeks after injection, with no significant differences between the two groups. Lips injected with hyaluronic acid filler of intermediate cross-link density resulted in more acute angles than lips injected with lightly cross-linked hyaluronic acid. The difference was statistically significant at each follow-up time point. No serious complications were observed throughout the study period. Our results imply that in patients who want a prominent upper lip lift, lip injections using hyaluronic acid fillers with intermediate cross-linking density can be a good option due to their lift capacity. The degree of cross-linking may not be a significant determinant of simple lip volume augmentation when other variables are constant.

## 1. Introduction

Lips are a key defining feature of the face and play an important role in expressing one’s age, attractiveness, and health [1]. Prominent signs of labial aging include loss of volume or lip thinning, increased appearance of vertical lines and wrinkles, and uneven texture, and they have a strong inverse relationship with perceived attractiveness [2]. Injectable fillers have been used to improve the appearance of aging lips, and hyaluronic acid (HA)-based dermal fillers are the most used products in lip enhancement.

HA has excellent biocompatibility, but it is a soluble polymer that rapidly degrades when injected into normal skin. Therefore, to provide the ability to fill and lift the skin, it is required to chemically modify the HA molecules using cross-linking, with an improvement of their mechanical properties and residence time at the implant site [3]. Theoretically, the strength of a HA gel increases gradually as the cross-linking density increases [4].

The corrective needs of a patient can range from a subtle enhancement of an already good shape and volume to a more comprehensive recontouring [5], including highlighting the vermilion borders, correcting the descending lip corners, and lifting the upper lip. Among the various aesthetic needs, particular interest lies in the nasolabial region. Changing the columella–labial relationships through various surgical procedures can give patients a youthful appearance by creating aesthetically pleasing local alterations [6]. The present study aimed to determine whether the mechanical properties and clinical efficacy are related to the cross-linking degree of HA filler products for lip augmentation.

## 2. Materials and Methods

### 2.1. Study Population

Eligible patients for the study were Korean female volunteers older than 18 who were willing to receive non-surgical lip augmentation using HA fillers. Exclusion criteria were (1) a history of systemic disease which may affect filler injection, (2) pregnancy or nursing, (3) anticoagulative medication, (4) allergies to HA or lidocaine, and (5) a history of prior filler injection of the lips within the past two years. Thirty-six patients meeting the inclusion criteria were enrolled in the study. The study protocol was set following the ethical standards of the institutional and national research committee (KGCP-2021-0910) and the ethical principles expressed in the Declaration of Helsinki.

### 2.2. Randomization

Using an online resource (https://www.randomizer.org/ (accessed on 14 September 2021)), developed for computer-generated randomization, the patients were assigned into two groups according to the injected HA filler product. Two different HA fillers (L’ORIENT NO.2 for Group A and L’ORIENT NO.4 for Group B) from a single manufacturer (Joonghun Pharmaceutical, Seoul, Korea) were used in this study (Table 1). Two products differ from each other in the degree of cross-linking with 1,4-butanediol diglycidyl ether (BDDE), where the cross-link density is low (1–2%) in L’ORIENT NO.2 and intermediate (2–3%) in L’ORIENT NO.4. Both products contain 20 mg/mL of HA and 0.3% lidocaine. The subjects were unaware of the filler product they received and were withheld from the information before the end of the study. As commercial products were used in the study, the injectors were not blinded to the products they were injecting.

### 2.3. Injection Protocol

Written informed consent was obtained from all patients after a detailed explanation of the technique, results, complications, and other treatment options. All patients were treated with the same injection protocol. After removing facial and lip makeups, a topical anesthetic cream (eutectic mix of 5% lidocaine and prilocaine base) was applied for 20 min under occlusion. Treatment areas were made aseptic with chlorhexidine before injection. Upon the patient’s supine position, either L’ORIENT NO.2 or L’ORIENT NO.4 filler product was injected using a standard technique described in the literature [7].

The technique involves (1) inserting the 27-gauge needle at the lateral edge of the lower lip, (2) tunneling the needle along the vermilion border to the midpoint of the lower lip, and (3) injecting the filler in a linear retrograde fashion. The product was injected along the body of the lower lip to achieve the desired volume augmentation. For the upper lip, the vermilion border was treated first to define the Cupid’s bow and a vermilion border before volumizing the upper lip body. The needle was inserted at the lateral edge of the upper lip and tunneled lateral to medial along the vermilion border to the Glogau-Klein points in the Cupid’s bow. A small amount of filler was placed into the Cupid’s bow to define this area. The remainder of the product was then injected in a retrograde fashion as the needle was withdrawn. For each patient, a total of 1.0 mL of the product was injected into the lips (0.4 mL in the upper lip and 0.6 mL in the lower lip). No touch-up injections were provided before 12 weeks post injection.

### 2.4. Follow-Up and Evaluation

The patients were followed at four and 12 weeks following the filler injection. Clinical efficacy was evaluated by blinded evaluators using the static photonumeric lip fullness scale (LFS) [8] and Global Aesthetic Improvement Scale (GAIS; –1, worsened; 0, unaltered; 1, improved; 2, much improved; 3, exceptionally improved), based on clinical photographs. Patients were also assessed to evaluate their satisfaction regarding volume, shape, attractiveness, and naturalness of the lips (0, not satisfied; 1, minimally satisfied; 2, satisfied; 3, very satisfied). At each follow-up visit, the patients were questioned regarding any adverse events.

Three-dimensional (3-D) photographs of the patient’s face were taken before the procedure and at each follow-up. The 3-D imaging system used in this study was VECTRA M3 (Canfield Scientific, Parsippany-Troy Hills, NJ, USA). The accuracy and reproducibility of results obtained using this system have been demonstrated in many works [9,10,11,12]. This system has also been used to measure volumetric changes after lip filler injections [13,14]. Care was taken to ensure a nonsmiling facial tone in pretreatment and post-treatment photographs.

After registration of each image to the pretreatment image, a closed area selection lined by vermilion borders was cropped from the original 3-D image (Figure 1) and presented to volume measurements. Volume measurements were made using a 3-D image analysis software (VECTRA Analysis Module, Canfield Scientific, USA) that compared the volume difference between the pretreatment and post-treatment images at each follow-up point in the *lips proper*. All volume measurements were recorded in milliliters.

The columella–labial angle (CLA), determined by the angle between the line from the subnasale to the labrale superius and the line from the subnasale to the most inferior columella (Figure 2), was measured and analyzed by comparing preoperative and postoperative 3-D photographs using the same software. To assess lip projection, we also measured the linear distance (in millimeters) from a reference plane (Rickett’s line or “E-line”) to the most protruding point of the upper and lower lip vermillion border.

Repeated measures analysis of variance (ANOVA) was used to compare measured values at each three time points. A paired *t*-test was used to evaluate the significance of the measurement changes at two follow-ups and between the groups. *p* < 0.05 was considered significant.

## 3. Results

Thirty-six patients were included in the study. Two patients (one from Group A and one from Group B) lost follow-up due to COVID-19 infection and were excluded from the final analysis. The mean age of the participants was 29.2 (range, 21–43 years) in Group A and 29.3 (range, 20–47 years) in Group B. Body mass index (BMI) of the subjects was not calculated since we did not measure the patients’ weight and height. One patient from Group B had a history of RESTYLANE (Galderma, Switzerland) injected in the lips, four years before the study. Acute adverse events include two cases of bruise in Group B and one case of local swelling in Group A, which were temporary and resolved within a week without intervention.

### 3.1. Non-Quantitative Evaluations

Treated patients achieved a statistically significant increase at four weeks from baseline in the LFS scores of both the upper and the lower lips. The mean LFS score at 12 weeks also showed a statistically significant increase from baseline. In both groups, the mean LFS scores at 12 weeks were lower than those at four weeks, and the difference was statistically significant (Table 2). GAIS scores and patients’ overall subjective satisfaction scores showed a similar tendency (Figure 3). Patients’ overall satisfaction scores were high at each follow-up, with no significant difference in both groups (2.8 and 2.7 at four weeks, *p* = 0.33; 2.6 and 2.6 at 12 weeks, *p* = 1.00). Of note, at 4 weeks, patients in Group A reported a higher satisfaction score than Group B regarding the naturalness of clinical results (2.9 and 2.6, *p* = 0.029). Nevertheless, none of our patients was noted to have an overcorrected appearance that requires the use of hyaluronidase at any point in the study.

### 3.2. Lip Volume Changes

Volume measurements of 3-D lip surface selections revealed a statistically significant increase in mean lip volume at four weeks and 12 weeks after injection, compared to the pretreatment volumes (3.1, 4.1, and 3.8 mL, each; *p* < 0.00001). At four weeks and 12 weeks after injection, lip volume increase in Group B was slightly higher than that of Group A, but the difference was not statistically significant. There was a tendency for the peak volume increase seen at four weeks following injection, but most patients retained a significant portion of their initial gain in lip volume at the 12-week time point (Table 3).

### 3.3. Lip Projection

Using the Rickett’s E-line as a reference plane, upper lip projection significantly increased at 4- and 12-weeks compared with baseline (*p* = 0.00011). The distance between the most protruding upper lip point and the reference line decreased from 1.5 mm at baseline to 0.6 mm at four weeks and 1.0 mm at 12 weeks, thus indicating increased projection of the upper lip during the follow-up period. Lower lip also showed increased projection (data not shown). The average increase in lip projection at each follow-up was slightly higher in Group B than Group A; however, the difference was statistically not significant (Table 3).

### 3.4. Columella-Labial Angle Changes

The mean CLA of the patients before the injection of a HA filler was 103.8 degrees. After four weeks of the injection, the mean CLA was 100.1 degrees. After 12 weeks, CLA decreased further to 99.5 degrees, indicating that the injection of the lips using HA fillers leads to a decrease in CLA by an average of 4.3 degrees. Subgroup analysis revealed a significant decrease in CLA at four weeks and 12 weeks, compared to the pretreatment values, in Group B (*p* = 0.000013), whereas the decrease was not significant in Group A, either at four weeks or 12 weeks after injection (*p* = 0.26035). The results of this variable are shown in Table 4 and Figure 4.

## 4. Discussion

In our study, we measured and analyzed the changes of CLA, regarded as one of the most important parameters to determine the aesthetics of the nasolabial region. The nose–lip–chin relationships are exceedingly important in determining facial aesthetics [15]. Although often measured differently by various methods, the most frequently used soft tissue parameter in orthodontic diagnosis is the CLA, a more “soft-tissue-oriented” version of the nasolabial angle (NLA) [16], which is formed by a line from the lower border of the nose to one representing the inclination of the upper lip [15]. Sutton et al. [17] have described this angle as increasing with age due to the decreased support and projection of the soft tissue envelope, leading to the loss of vermilion. Obtuse CLA, in addition to the decrease in facial fat, is regarded to create the typical appearance of the aging face. A more pleasant and youthful appearance can be appreciated when the distance from the nose to the vermilion border of the upper lip shortens and the visualized quantity of the lip vermilion increases [6], which can be achieved by injecting dermal fillers in the upper lip. Interestingly, Ghannam and Bageorgou [18] recently found an average of a 5.1% reduction in the philtrum length in patients who received more than one HA filler injection in the upper lip, which implies that fillers injected in the upper lip can result in similar clinical effects of surgical lip lift procedures.

In the present study, we demonstrated that the injection of the lips using cross-linked HA fillers leads to a significant decrease in CLA, in line with a previous report [19]. After subgroup analysis, we further demonstrated that the post-treatment decrease in CLA was statistically significant only in Group B but not in Group A at both 4-week and 12-week follow-ups. CLA or NLA with the best esthetics generally shows a relatively wide range when the degree of upper lip protrusion is small [20]. However, when lip protrusion is marked, the aesthetically preferred CLA tends to fall within a narrow range, with a strong preference for more acute CLA related to the degree of upper lip protrusion [20]. Considering that surgical lip lift results in a significant decrease in the angle of the nasolabial region [21], a decrease in CLA found in our study implies the lip lifting effect of HA filler injections. Furthermore, HA fillers with higher cross-linking densities may provide a more lip lift, as shown in Group B, in which the CLA decrease was statistically significant. However, it should also be noted that attractive CLA is greatly affected by the projection of the nasal tip. Since the nasal tip is often under-projected in the Asian population, surgical rhinoplasty or nasal filler injections, focusing on the nasal tip and the columella, may produce a more youthful CLA when performed in conjunction with lip filler injections.

Lifting capacity can be defined as the capacity to lift tissue and resist deformation after the injection. A liquid, or a weak gel, will not resist deformation and will therefore displace in the direction of least resistance, and the desired tissue lift will be achieved to less extent [22]. Along with gel cohesivity, elastic modulus (G′) is viewed as a general indicator of the lifting capacity of HA fillers [23]. G′ is a quantifiable property of the gel, describing its ability to resist deformation [22]. The degree of modification can significantly affect HA gel strength. As the cross-link density of a gel increases, the distance between the cross-linked segments becomes shorter. When a load is applied, these shorter segments require a greater force to deflect. Thus, increasing cross-link density strengthens the overall network, thereby increasing G′ or the hardness of the gel [3].

In this regard, it is plausible to think that in our study, the filler Product B (L’ORIENT NO.4) may have a more lip lifting capacity than Product A (L’ORIENT NO.2) since among fillers with the same composition and cross-linking technology, G′ has a positive correlation to overall lift capacity [23]. The results of our study revealed that a HA filler product with an intermediate degree of cross-linking showed a distinct upper lip lift effect than a lightly cross-linked HA filler product (Figure 5 and Figure 6). However, it should be kept in mind that lift does not always correlate with increasing G′ because many other parameters also influence performance when comparing fillers with different compositions or different cross-linking technologies [23]. It is noteworthy that patients’ satisfaction scores regarding natural-looking results were higher in Group A than in Group B, suggesting the use of a HA filler product with higher lifting capacity may not always be suitable for lip augmentation in a certain group of patients, e.g., conservative patients and the elderly. Using lightly cross-linked HA fillers with less lifting capacity may suit this subgroup of patients well.

Most studies addressing the efficacy and longevity of lip filler injection have used subjective evaluation measures such as patient satisfaction surveys and physician observation [5,24]. Validated photonumeric lip fullness scales have also been used to objectively compare lip augmentation results in several clinical studies [24,25,26,27]. However, only a few studies [13,14] have used quantitative measurements to assess the volume enhancement effects of lip filler injections. In the present study, we used 3-D stereophotogrammetry to provide objective data regarding the augmentation efficacy of injectable cross-linked HA gel as a lip volumizing filler. In addition to providing objectivity in evaluating volumetric enhancements, 3-D imaging is a useful tool to measure and assess linear and angular variables in facial aesthetic surgery. The 3-D stereophotogrammetry system used in our study (VECTRA M3) has been reported to be repeatable, giving accurate measures within the references established in several studies [9,10], with key facial angles presenting excellent intraclass correlation coefficient values [11,12]. However, it should be noted that the 3-D volume estimation in the study cannot be fully supported by precision in reproducibility since this system reads and interpolate the 3-D mesh data to calculate the theoretical surface volume of the closed area, instead of measuring the real volume of the lips.

As expected, the 3-D measurement in our study revealed a statistically significant increase in lip volume following the treatment (mean 1.0 mL increase at four weeks, mean 0.7 mL increase at 12 weeks). The result is in line with a study by Nikolis et al. [14], who demonstrated the mean changes from baseline in total lip volume and surface area increased significantly (a mean 0.53 mL increase at eight weeks) following the injection of RESTYLANE KYSSE and RESTYLANE REFYNE (Galderma, Lausanne, Switzerland), using the same 3-D photography system as the present study. Considering that the injected volume in a study by Nikolis et al. [14] (1.6–1.8 mL) was larger than that of the present study (1.0 mL), the HA filler products used in our study (L’ORIENT NO.2 and L’ORIENT NO.4) seem to have a very strong volume-enhancing property when used in the lips. Moreover, lip volume was still significantly increased compared with baseline at the 12-week follow-up, with no statistical difference between the two product groups in our study. This tendency was also found in LFS, GAIS, and patients’ overall subjective satisfaction scores. Our study also has several limitations. The medium-term nature of the follow-up (12 weeks) makes the present study difficult to demonstrate any long-term results, with the only temporary improvement of CLA. A relatively small number of patients may limit the statistical power of our results. The study population was limited to Korean females in a relatively younger age range. Future studies with more participants with a wide age range and different ethnicity or gender, are suggested to validate our findings further.

Another limitation of this study is that the static nature of assessment. The lips and the perioral area are highly mobile and can be more properly evaluated with dynamic evaluation, as demonstrated by a recent study [14], which measured the “stretch” of a 3-D perioral surface. However, this study could not compare the perioral dynamics between two filler products with different rheological properties. Considering that dynamic facial changes that may occur using a high G′ filler are sometimes discouraged in its use on the lips, a further evaluation of perioral dynamism according to a specific filler product can provide more clinical implications in producing natural-appearing results. Future studies may compare 3-D volumetric changes with qualitative and dynamic changes in lip aesthetics.

## 5. Conclusions

An elastic and cohesive HA filler with intermediate cross-link density can be used to create attractive and younger-looking lips, taking advantage of the lifting capacity and persistence of the gel in place. Patients who want more natural results may be better indicated using lightly cross-linked HA gels in lip augmentation.

## Figures and Tables

**Figure 1 jcm-11-04554-f001:**
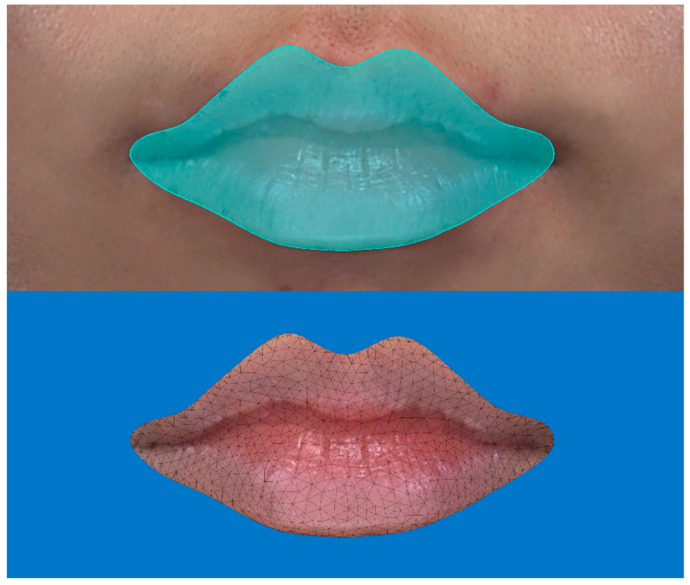
A closed area selection of the lips, cropped from the original 3-dimensional image.

**Figure 2 jcm-11-04554-f002:**
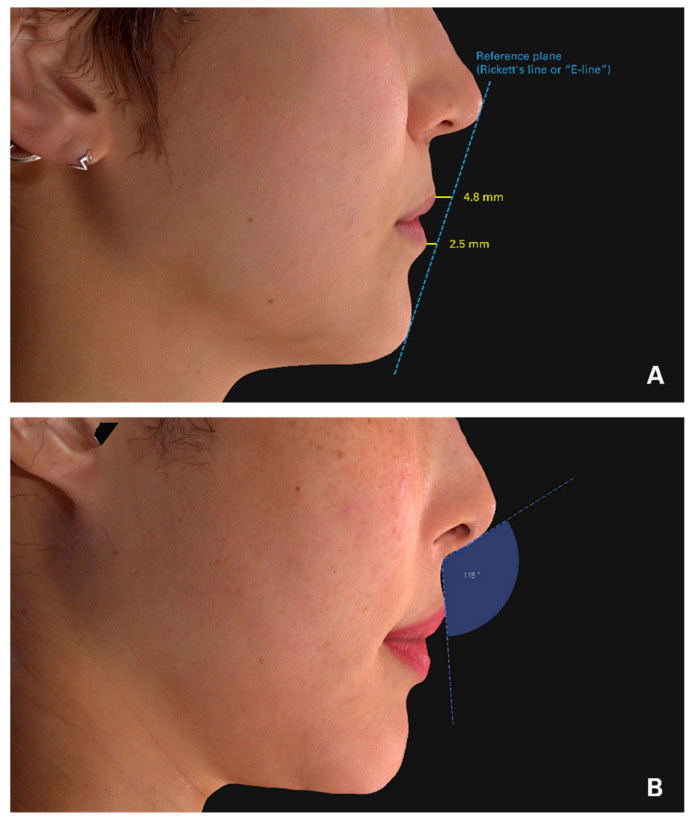
Photographic illustrations show linear and angular measurements used in the study. (**A**). The lip projection was determined in relation to Rickett’s E-line. (**B**). An example of the columella–labial angle measurement.

**Figure 3 jcm-11-04554-f003:**
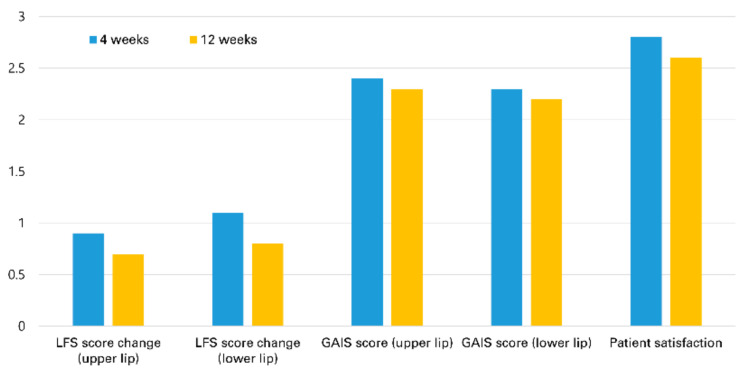
Lip Fullness Scale (LFS) score changes, Global Aesthetic Improvement (GAIS) scores, and patient satisfaction scores at 4 and 12 weeks after lip filler injection.

**Figure 4 jcm-11-04554-f004:**
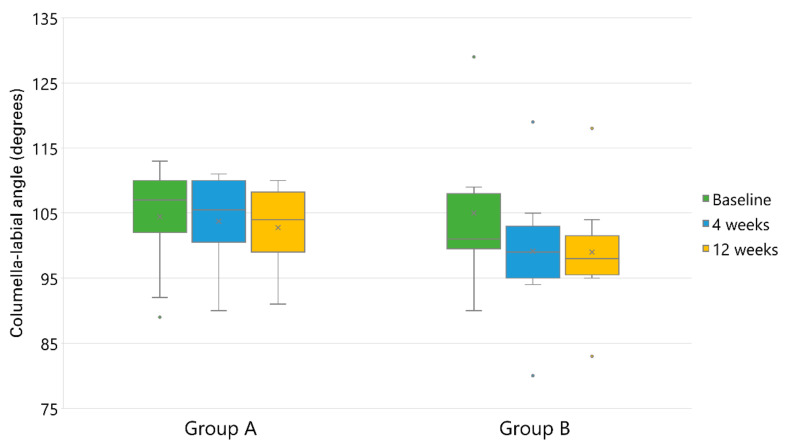
A decrease of columella–labial angle after lip injections with hyaluronic acid fillers with different degrees of cross-linking (Group A, light cross-linking; Group B, intermediate cross-linking). × represents the mean. Dots represent outliers.

**Figure 5 jcm-11-04554-f005:**
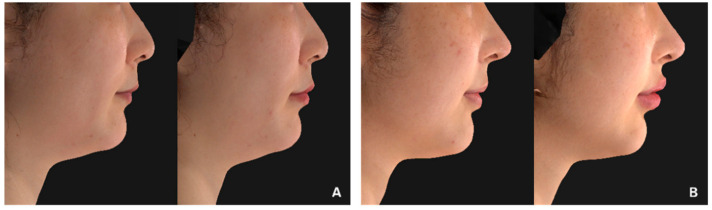
Profile views of patients injected with 1.0 mL of hyaluronic acid fillers in the lips, before and 12 weeks after treatment. (**A**). Hyaluronic acid filler with a low degree of cross-linking (patient 4). (**B**). Hyaluronic acid filler with an intermediate degree of cross-linking (patient 11).

**Figure 6 jcm-11-04554-f006:**
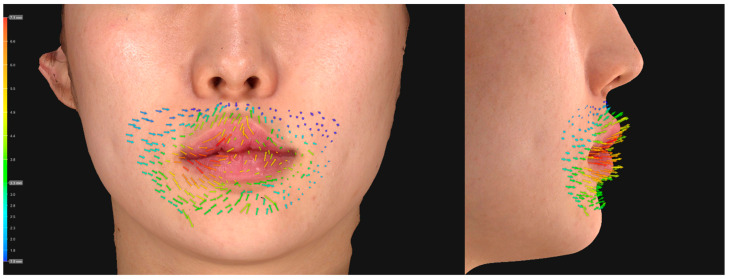
Visualization of an upper lip lift after injection of a cross-linked hyaluronic acid filler in a Group B patient. The difference in skin displacement between the 3-dimensional images taken at baseline and 12 weeks after injection was represented through vectors.

**Table 1 jcm-11-04554-t001:** Properties of hyaluronic acid filler products used in the study.

	L’ORIENT NO.2	L’ORIENT NO.4
Designated group	Group A	Group B
Total HA ^1^ concentration (mg/mL)	20	20
Cross-linking agent	BDDE ^2^	BDDE
Degree of cross-linking	1–2%	2–3%
Elastic modulus, G′ (Pa)	249	436
Viscous modulus, G′′ (Pa)	34.67	43.70
Tan δ (G′′/G′)	0.14	0.10
Cohesiveness (N)	−0.21	−0.21

^1^ hyaluronic acid; ^2^ 1,4-butanediol diglycidyl ether.

**Table 2 jcm-11-04554-t002:** Lip fullness scale (0–4; mean ± standard deviation) by blinded evaluators at each time point.

	Pretreatment	4 Weeks	12 Weeks	*p*-Value
Upper lip				
All patients	2.1 ± 0.54	3.0 ± 0.51	2.8 ± 0.49	<0.00001 *
Group A	2.1 ± 0.66	2.9 ± 0.53	2.6 ± 0.45	<0.00001 *
Group B	2.2 ± 0.44	3.1 ± 0.47	3.0 ± 0.43	<0.00001 *
Lower lip				
All patients	2.0 ± 0.48	3.1 ± 0.47	2.8 ± 0.49	<0.00001 *
Group A	2.0 ± 0.58	3.0 ± 0.44	2.7 ± 0.48	<0.00001 *
Group B	2.0 ± 0.41	3.1 ± 0.49	3.0 ± 0.47	<0.00001 *

Repeated measures analysis of variance (ANOVA) was used for statistical analysis; * Statistically significant.

**Table 3 jcm-11-04554-t003:** Increase in the lip volume and the upper lip projection at 4- and 12-week time points.

	Group A	Group B	*p*-Value
Total lip volume change (mL)			
4 weeks	0.8 ± 0.45	1.1 ± 0.81	0.1371
12 weeks	0.6 ± 0.56	0.8 ± 0.56	0.3832
Anterior displacement of the upper lip (mm)			
4 weeks	0.7 ± 1.12	1.1 ± 0.81	0.2383
12 weeks	0.3 ± 0.78	0.7 ± 1.09	0.2402
Anterior displacement of the lower lip (mm)			
4 weeks	0.9 ± 1.25	0.8 ± 0.89	0.8108
12 weeks	0.8 ± 0.90	0.8 ± 0.90	1

A paired *t*-test was used for comparison between two groups.

**Table 4 jcm-11-04554-t004:** The columella–labial angle (degrees; mean ± standard deviation) measured at each follow-up.

	Pretreatment	4 Weeks	12 Weeks	*p*-Value
All patients	103.8 ± 9.50	100.1 ± 9.99	99.5 ± 9.33	0.00004 *
Group A	104.4 ± 7.46	103.8 ± 7.35	102.8 ± 6.68	0.26035
Group B	103.2 ± 11.50	96.4 ± 11.19	96.3 ± 10.70	0.000013 *

Repeated measures analysis of variance (ANOVA) was used for statistical analysis; * Statistically significant.

## Data Availability

Not applicable.

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
