# Peer review of "Lip Lifting Efficacy of Hyaluronic Acid Filler Injections: A Quantitative Assessment Using 3-Dimensional Photography"

_jcm, 2022, doi:10.3390/jcm11154554_

Round 1

Reviewer 1 Report

Dear authors i have read your research with interest. From a formal scientic point of view the work is well executed, with a good and reproducible analysis. Nontheless, I have few queries to submit:

1) The aim of this reasearch is if cross-linking degree is somewhat related to other clinical effects other than resorption rate, and so I have some concerns on the small follow-up presented. This is indeed a limitation of the study that was explicited in the end of the discussion section.

2) You mentionend "dynamic changes" at the end of the discussion section and in conclusion section you said "Patients who want more natural results may be better indicated using lightly cross-linked HA gels in lip augmentation." I agree with the last sentence. This study is mostly a reseach on static HA fillers features. I suggest to add in the discussion section a paragraph on dynamic changes/features that may occur using an high G' filler, since its use is usually discouraged due to possible visible granules, asymmetries and less natural appearence in dynamic movements.

3) line 156 add word "significant" at the end of the sentence.

Best Regards

Author Response

We would like to thank the reviewers for their thoughtful comments and efforts towards improving our manuscript. In the following, we present our response specific to each reviewer comments.

[Reviewer’s comment] 1) The aim of this reasearch is if cross-linking degree is somewhat related to other clinical effects other than resorption rate, and so I have some concerns on the small follow-up presented. This is indeed a limitation of the study that was explicited in the end of the discussion section.

[Authors’ response] Thank you very much for your feedback. As the reviewer clearly mentioned, the effect of HA cross-linking can best be compared under a long-term clinical monitoring setup. We admit that this is really the main weakness of the present study.

[Reviewer’s comment] 2) You mentionend "dynamic changes" at the end of the discussion section and in conclusion section you said "Patients who want more natural results may be better indicated using lightly cross-linked HA gels in lip augmentation." I agree with the last sentence. This study is mostly a reseach on static HA fillers features. I suggest to add in the discussion section a paragraph on dynamic changes/features that may occur using an high G' filler, since its use is usually discouraged due to possible visible granules, asymmetries and less natural appearence in dynamic movements.

[Authors’ response] We appreciate the reviewer’s valuable suggestion. We put emphasizing phrases regarding the dynamic facial changes after lip filler injections: “Another limitation of this study is that the static nature of assessment. quantitative changes are not always correlated with qualitative measures of volume change. The lips and the perioral area are highly mobile and can be more properly evaluated with dynamic evaluation, as demonstrated by a recent study [14], which measured the "stretch" of a 3-D perioral surface. However, this study could not compare the perioral dynamics between two filler products with different rheologic properties. Considering that dynamic facial changes that may occur using a high G′ filler are sometimes discouraged in its use on the lips, a further evaluation of perioral dynamism according to a specific filler product can provide more clinical implications in producing natural-appearing results. Future studies may compare 3-D volumetric changes with qualitative and dynamic changes in lip aesthetics.”

[Reviewer’s comment] 3) line 156 add word "significant" at the end of the sentence.

[Authors’ response] Thank you very much for the suggestion. The word “significant” is now present at the end of the sentence mentioned.

Due to many changes made in the manuscript, we have revised the Abstract to conform to the revision.

Again, we would like to show our deepest appreciation regarding the reviewer’s valuable comments and suggestions.

Nark-Kyoung Rho (the corresponding author)

Reviewer 2 Report

Summary:

In this manuscript, the authors describe the effect of different cross-linking injectable hyaluronic acid gels in terms of volume effect and change in nasolabial angle using 3-D surface scanning.

The authors present an interesting approach to compare two hyaluronic acid fillers not used in my country using subjective and objective methods.

The illustrations are purposeful. The introduction and methodology are clear with minor weaknesses. The statistical evaluation seems to me partly not secure enough and needs a revision from my point of view. The 3D volume measurement seems questionable to me, not because of 3D and the technique, but the choice of how volumes were measured (see references of similar publications). It is a pity that no other 3-D measurements like projection (see references) were calculated, which besides volume could have given the readers even more information about the effect. In the results section, I feel there is a lack of correct quantification of statistical values and thus significance. There is a discrepancy between text, tables and figures. The multiple influencing factors on rheology of hyaluronic acid fillers including G-Prime, which are CORRECTLY mentioned in the discussion, do not find a place in most of the paper and in the title of the manuscript. Perhaps the title should be reconsidered or the focus of "crosslinking" toned down a bit.

Introduction:

Useful introduction to the research question of the student research paper. However, few references regarding current literature, but this seems to be made up for in the discussion.

Materials and Methods:

2.1 Study population:

The listing (1 - 6) of exclusion criteria does not include no. 2. Either former no. 2 was deleted in the course of manuscript revision or it was counted incorrectly.

In particular, exclusion criterion No. 6 with no filler application for at least 2 years does not appear to be well founded. Studies could show that even after years fillers could still be detected in the MRI. Thus, the question remains - were there never treated patients or were there patients with pre-treatment > 2 years? This otherwise needs to be addressed in the discussion.

The authors state that "thirty-five patients" met the inclusion criteria. However, the study states 36 subjects in remaining parts.

2.2 Randomization:

The section with randomization is relatively short for emphasizing it. How was the randomization done? Were the study participants aware of which product they were receiving? Please provide more details.

If you read the study reference 17 by Nikolis et al you will see that a volume change was measured pre-therapeutically to post-therapeutically via surface / volume change. From the available manuscript it appears that an "current volume" state of lip volume was measured using a similar marked area as in Nikolis et al. However, correct volume values cannot be reproducibly measured with surface measurements as this involves interpolation of the posterior wall which is inaccurate and may be different/re-set even in the same individual over multiple time periods.

Although I am a proponent of 3D, I view the measurement of 3D lip volume critically in terms of implementation.

Based on the collected 3D scans, a direct volume change should be measurable and more meaningful than the "theoretical" and "relative" volume states.

In addition, it is a pity that no other 3D data were used. Here there would be e.g. "surface area" expansion, dimensions of the lip, projection of the lips, etc.

3. Results

Two patients are unfortunately not available for the final analysis due to Covid-19. Which patients from which group are these?

Since weight and height are relatively important for "randomization" and "comparative studies" but are missing in the manuscript, they should also be mentioned.

"Adverse events“ are mentioned but without differentiation in which group they occurred.

3.2 Lip Volume Change

Here I refer again to the not meaningful volume measurement of an "current volume" state of the lips. More meaningful is the post-pre change between the 3 time points.

Statistical Testing

In addition, the paired T-test does not seem adequate for 3 time points. Repeated measures ANOVA would be more correct per group. This applies to all paired comparisons in this manuscript.

While the manuscript emphasizes a „comparison“ between the two products, in the results each group is assessed basically for the treatment effect that something happened.

Although one can argue that this is more than enough, the „comparison“ between both groups is subjective and not statistically proven. One could measure the difference between treatments (delta) from group A and B.

"Non-significant" results are described without a P value.

Figure 4 describes "P < 0.05" plotted over Group B but the text describes "P < 0.01".

Entering the values from Table 4 into a simple statistical tool with N = 18 or N = 16 (if 2 participants dropped out because of Covid), there appears to be no statistically significant differences for Group B based on the mean and standard deviation which is given. I did this rather unpleasant test because the delta between the measurement pairs seemed very small to me with a basic value of almost 100 of the mean values with similar standard deviation.

I think a simple box plot might give some clarity compared to Figure 4 and the given values in Figure 4, the text and my concerns.

Author Response

We would like to thank the reviewers for their thoughtful comments and efforts towards improving our manuscript. In the following, we present our response specific to each reviewer comments.

[Reviewer’s comment] The authors present an interesting approach to compare two hyaluronic acid fillers not used in my country using subjective and objective methods. The illustrations are purposeful. The introduction and methodology are clear with minor weaknesses.

[Authors’ response] Thank you very much for your favorable feedback.

[Reviewer’s comment] The statistical evaluation seems to me partly not secure enough and needs a revision from my point of view.

[Authors’ response] Thank you for the feedback. We understand that a more sophisticated statistical analysis may help determine whether our findings can be further justified. Please note the specific answers and the corresponding changes in response to the separate comments.

[Reviewer’s comment] The 3D volume measurement seems questionable to me, not because of 3D and the technique, but the choice of how volumes were measured (see references of similar publications). It is a pity that no other 3-D measurements like projection (see references) were calculated, which besides volume could have given the readers even more information about the effect.

[Authors’ response] We appreciate the reviewer’s valuable comment. We agree that incorporating other linear measurements, such as lip projection, into the 3-D volume measurement can provide further information to the readers. Please note the specific answers and the corresponding changes in response to the separate comments.

[Reviewer’s comment] In the results section, I feel there is a lack of correct quantification of statistical values and thus significance. There is a discrepancy between text, tables and figures.

[Authors’ response] We deeply apologize for our technical mistakes in the preparation of the manuscript regarding the inconsistency of the values in the main text, tables, and figures, which must be edited correctly. Please note the specific answers and the corresponding changes in response to the separate comments.

[Reviewer’s comment] The multiple influencing factors on rheology of hyaluronic acid fillers including G-Prime, which are CORRECTLY mentioned in the discussion, do not find a place in most of the paper and in the title of the manuscript. Perhaps the title should be reconsidered or the focus of "crosslinking" toned down a bit.

[Authors’ response] The authors discussed on changing the title of the manuscript, as carefully suggested by the reviewer. The revised title is as follows: Lip Lifting Efficacy of Hyaluronic Acid Filler Injections: A Quantitative Assessment Using 3-Dimensional Photography

[Reviewer’s comment] Introduction: Useful introduction to the research question of the student research paper. However, few references regarding current literature, but this seems to be made up for in the discussion.

[Authors’ response] The authors appreciate the reviewer’s comment.

[Reviewer’s comment] Materials and Methods: 2.1 Study population: The listing (1 - 6) of exclusion criteria does not include no. 2. Either former no. 2 was deleted in the course of manuscript revision or it was counted incorrectly.

[Authors’ response] We deeply apologize for our careless mistakes which have not been corrected before submitting the manuscript for peer-review. The listing of exclusion criteria has been changed correctly (1, 2, 3, 4, 5).

[Reviewer’s comment] In particular, exclusion criterion No. 6 with no filler application for at least 2 years does not appear to be well founded. Studies could show that even after years fillers could still be detected in the MRI. Thus, the question remains - were there never treated patients or were there patients with pre-treatment > 2 years? This otherwise needs to be addressed in the discussion.

[Authors’ response] We appreciate the reviewer’s comment, which the authors did not put appropriate consideration during the preparation of the manuscript. To answer the reviewer’s question, we telephoned the study subjects and found out that there was one patient who had a history of lip filler injection several years prior to the study. We included the information in the first paragraph of the result section: “One patient from Group B had a history of RESTYLANE (Galderma, Switzerland) injected in the lips, four years before the study.”

[Reviewer’s comment] The authors state that "thirty-five patients" met the inclusion criteria. However, the study states 36 subjects in remaining parts.

[Authors’ response] Again, we apologize for not correcting the technical errors before submission. We correctly changed the number of enrolled patients: “Thirty-six patients meeting the inclusion criteria were enrolled in the study.”

[Reviewer’s comment] 2.2 Randomization: The section with randomization is relatively short for emphasizing it. How was the randomization done? Were the study participants aware of which product they were receiving? Please provide more details.

[Authors’ response] The patients were randomized according to an online computer-generated randomization resource, and they were unaware of the product they received. The following sentences were included in the “randomization” paragraph: “Using an online resource (https://www.randomizer.org/), developed for computer-generated randomization, the patients were randomized assigned into two groups (18 each) according to the injected HA filler product. …… The subjects were unaware of the filler product they received and were withheld from the information before the end of the study. As commercial products were used in the study, the injectors were not blinded to the products they were injecting.”

[Reviewer’s comment] If you read the study reference 17 by Nikolis et al you will see that a volume change was measured pre-therapeutically to post-therapeutically via surface / volume change. From the available manuscript it appears that an "current volume" state of lip volume was measured using a similar marked area as in Nikolis et al. However, correct volume values cannot be reproducibly measured with surface measurements as this involves interpolation of the posterior wall which is inaccurate and may be different/re-set even in the same individual over multiple time periods. Although I am a proponent of 3D, I view the measurement of 3D lip volume critically in terms of implementation. Based on the collected 3D scans, a direct volume change should be measurable and more meaningful than the "theoretical" and "relative" volume states.

[Authors’ response] Thank you very much for your valuable criticism of the research methodology. We admit that the current 3-D surface volume measurement technology has limitations in reproducibility, as the reviewer commented. We included the phrase on this limitation in the discussion: “However, it should be noted that the 3-D volume estimation in the study cannot be fully supported by precision in reproducibility since this system reads and interpolate the 3-D mesh data to calculate the theoretical surface volume of the closed area, instead of measuring the real volume of the lips.”

[Reviewer’s comment] In addition, it is a pity that no other 3D data were used. Here there would be e.g. "surface area" expansion, dimensions of the lip, projection of the lips, etc.

[Authors’ response] Because we used the 3-D mesh data for analysis, it was possible for us to retrospectively perform additional 3-D measurements recommended by the reviewer. Specifically, we additionally measured the “lip projections” of each three time points, as the reviewer strongly recommended, using the “E-line” as a reference plane. This additional measurement has been included in the materials and methods section: “To assess lip projection, we also measured the linear distance (in millimeters) from a reference plane (Rickett's line or “E-line”) to the most protruding point of the upper and lower lip vermillion border.” A new figure illustrating the lip projection measurement was added as Figure 2A. and the results were additionally presented in the new Table 3.

[Reviewer’s comment] 3. Results. Two patients are unfortunately not available for the final analysis due to Covid-19. Which patients from which group are these?

[Authors’ response] One from Group A and one from Group B, each. This information has been added in the first sentence of the first paragraph of Results section.

[Reviewer’s comment] Since weight and height are relatively important for "randomization" and "comparative studies" but are missing in the manuscript, they should also be mentioned. Since "Adverse events“ are mentioned but without differentiation in which group they occurred.

[Authors’ response] We regret not checking the subjects’ weight and height information, which was briefly mentioned in the revised manuscript. Also, we added the adverse event information regarding which groups they developed.

[Reviewer’s comment] 3.2 Lip Volume Change. Here I refer again to the not meaningful volume measurement of an "current volume" state of the lips. More meaningful is the post-pre change between the 3 time points.

[Authors’ response] Thank you very much for the precious comment. We further analyzed the “changes” of values between time points. The results were presented in the new Table 3.

[Reviewer’s comment] Statistical Testing. In addition, the paired T-test does not seem adequate for 3 time points. Repeated measures ANOVA would be more correct per group. This applies to all paired comparisons in this manuscript. While the manuscript emphasizes a „comparison“ between the two products, in the results each group is assessed basically for the treatment effect that something happened.

[Authors’ response] We truly appreciate this comment. We changed the statistical analysis from paired t-test to repeated measures ANOVA to compare values at 3 different time points. The sentence in the materials methods (statistical analysis) was changed as follows: “Repeated measures analysis of variance (ANOVA) was used to compare measured values at each three time points. A paired t-test was used to evaluate the significance of the measurement changes at two follow-ups and between the groups.” Accordingly, the result tables (Tables 2 and 4) have been changed. Please refer to the revised manuscript for detailed changes.

[Reviewer’s comment] Although one can argue that this is more than enough, the „comparison“ between both groups is subjective and not statistically proven. One could measure the difference between treatments (delta) from group A and B.

[Authors’ response] As the reviewer suggested, we made a further comparison of the “difference” or “change” of the measured values between Groups A and B, at each follow-up time points (4- and 12-weeks). The results were presented in the new Table 3.

[Reviewer’s comment] "Non-significant" results are described without a P value. Figure 4 describes "P < 0.05" plotted over Group B but the text describes "P < 0.01".

[Authors’ response] Thank you very much for pointing out the detailed points. In the revised manuscript text, we corrected the mistake and put the exact P-values for each group.

[Reviewer’s comment] Entering the values from Table 4 into a simple statistical tool with N = 18 or N = 16 (if 2 participants dropped out because of Covid), there appears to be no statistically significant differences for Group B based on the mean and standard deviation which is given. I did this rather unpleasant test because the delta between the measurement pairs seemed very small to me with a basic value of almost 100 of the mean values with similar standard deviation. I think a simple box plot might give some clarity compared to Figure 4 and the given values in Figure 4, the text and my concerns.

[Authors’ response] We understand that the relatively small number of values of each group may jeopardize the results of statistical analysis. To visualize and clarify the mean and the standard deviation of each group, we changed the graph type of Figure 4 to a box-and-whisker plot, as the reviewer recommended.

Due to many changes made in the manuscript, we have revised the Abstract to conform to the revision.

Again, we would like to show our deepest appreciation regarding the reviewer’s valuable comments and suggestions.

Nark-Kyoung Rho (the corresponding author)

Reviewer 3 Report

This study is interesting and demonstrates using dermal fillers in the lip to obtain an improved CLA. The follow up is only 12 weeks but the authors note this as a limitation and it does not affect the concept illustrated, that dermal fillers can temporarily improve the CLA. The study population only included Korean females in a relatively narrow age range. This needs to be discussed as a limitation. Further, use of dermal filler to improve CLA should be discussed along with other procedures (i.e., rhinoplasty) that affect CLA. 

Author Response

We would like to thank the reviewers for their thoughtful comments and efforts towards improving our manuscript. In the following, we present our response specific to each reviewer comments.

[Reviewer’s comment] The follow up is only 12 weeks but the authors note this as a limitation and it does not affect the concept illustrated, that dermal fillers can temporarily improve the CLA.

[Authors’ response] Thank you very much for the comment. The lack of long-term follow-up is indeed the most serious weakness of the present study. We admit that the study result implies only the temporary improvement of CLA, as the reviewer pointed out.

[Reviewer’s comment] The study population only included Korean females in a relatively narrow age range. This needs to be discussed as a limitation.

[Authors’ response] We appreciate the reviewer’s comment. We added this mention in the discussion-limitations paragraph: “The study population was limited to Korean females in a relatively younger age range. Future studies with more participants with a wide age range and different ethnicity or gender, are suggested to validate our findings further.”

[Reviewer’s comment] Further, use of dermal filler to improve CLA should be discussed along with other procedures (i.e., rhinoplasty) that affect CLA.

[Authors’ response] Thank you for the valuable suggestion. We totally agree that lip filler injections can make the younger-looking CLA if they are combined with other aesthetic procedures of the nose, since the nasal tip is often under-projected in the Asian population. Surgical rhinoplasty or filler injections, focusing on the nasal tip and the columella, may produce a more youthful CLA if performed in conjunction with lip filler injections. In discussion, we included a short sentence addressing this topic: “However, it should also be noted that attractive CLA is greatly affected by the projection of the nasal tip. Since the nasal tip is often under-projected in the Asian population, surgical rhinoplasty or nasal filler injections, focusing on the nasal tip and the columella, may produce a more youthful CLA when performed in conjunction with lip filler injections.”

Due to many changes made in the manuscript, we have revised the Abstract to conform to the revision.

Again, we would like to show our deepest appreciation regarding the reviewer’s valuable comments and suggestions.

Nark-Kyoung Rho (the corresponding author)

Round 2

Reviewer 2 Report

The targeted revisions have been made to the extent recognizable.